# DISTANTLY SUPERVISED END-TO-END MEDICAL ENTITY EXTRACTION FROM ELECTRONIC HEALTH RECORDS WITH HUMAN-LEVEL QUALITY

## ABSTRACT

Medical entity extraction (EE) is a standard procedure used as a first stage in medical texts processing. Usually Medical EE is a two-step process: named entity recognition (NER) and named entity normalization (NEN). We propose a novel method of doing medical EE from electronic health records (EHR) as a single-step multi-label classification task by fine-tuning a transformer model pretrained on a large EHR dataset. Our model is trained end-to-end in an distantly supervised manner using targets automatically extracted from medical knowledge base. We show that our model learns to generalize for entities that are present frequently enough, achieving human-level classification quality for most frequent entities. Our work demonstrates that medical entity extraction can be done end-to-end without human supervision and with human quality given the availability of a large enough amount of unlabeled EHR and a medical knowledge base.

## 1 INTRODUCTION

Wide adoption of electronic health records (EHR) in the medical care industry has led to accumulation of large volumes of medical data (Pathak et al., 2013). This data contains information about the symptoms, syndromes, diseases, lab results, patient treatments and presents an important source of data for building various medical systems (Birkhead et al., 2015). Information extracted from medical records is used for clinical support systems (CSS) (Shao et al., 2016) (Topaz et al., 2016) (Zhang et al., 2014), lethality estimation (Jo et al., 2015) (Luo & Rumshisky, 2016), drug side-effects discovery (LePendu et al., 2012) (Li et al., 2014) (Wang et al., 2009), selection of patients for clinical and epidemiological studies (Mathias et al., 2012) (Kaelber et al., 2012) (Manion et al., 2012), medical knowledge discovery (Hanauer et al., 2014) (Jensen et al., 2012) and personalized medicine (Yu et al., 2019). Large volumes of medical text data and multiple applicable tasks determine the importance of accurate and efficient information extraction from EHR.

Information extraction from electronic health records is a difficult natural language processing task. EHR present a heterogeneous dynamic combination of structured, semi-structured and unstructured texts. Such records contain patients' complaints, anamneses, demographic data, lab results, instrumental results, diagnoses, drugs, dosages, medical procedures and other information contained in medical records (Wilcox, 2015). Electronic health records are characterised by several linguistic phenomena making them harder to process.

- Rich special terminology, complex and volatile sentence structure.
- Often missing term parts and punctuation.
- Many abbreviations, special symbols and punctuation marks.
- Context-dependant terms and large number of synonims.
- Multi-word terms, fragmented and non-contiguous terms.

From practical point of view the task of medical information extraction splits into entity extraction and relation extraction. We focus on medical entity extraction in this work. In the case of medical texts such entities represent symptoms, diagnoses, drug names etc.

Entity extraction, also referred as Concept Extraction is a task of extracting from free text a list of concepts or entities present. Often this task is combined with finding boundaries of extracted entities as an intermediate step. Medical entity extraction in practice divides into two sequential tasks: Named entity recognition (NER) and Named entity normalization (NEN). During NER sequences of tokens that contain entities are selected from original text. During NEN each sequence is linked with specific concepts from knowledge base (KB). We used Unified Medical Language System (UMLS) KB (Bodenreider, 2004) as the source of medical entities in this paper.

In this paper we make the following contributions. First, we show that a single transformer model (Devlin et al., 2018) is able to perform NER and NEN for electronic health records simultaneously by using the representation of EHR for a single multi-label classification task. Second, we show that provided a large enough number of examples such model can be trained using only automatically assigned labels from KB to generalize to unseen and difficult cases. Finally, we empirically estimate the number of examples needed to achieve human-quality medical entity extraction using such distantly-supervised setup.

## 2 RELATED WORK

First systems for named entity extraction from medical texts combined NER and NEN using term vocabularies and heuristic rules. One of the first such systems was the Linguistic String Project - Medical Language Processor, described in Sager et al. (1986). Columbia University developed Medical Language Extraction and Encoding System (MedLEE), using rule-based models at first and subsequently adding feature-based models (Friedman, 1997). Since 2000 the National Library of Medicine of USA develops the MetaMap system, based mainly on rule-based approaches (Aronson et al., 2000). Rule-based approaches depend heavily on volume and fullness of dictionaries and number of applied rules. These systems are also very brittle in the sense that their quality drops sharply when applied to texts from new subdomains or new institutions.

Entity extraction in general falls into three broad categories: rule-based, feature-based and deep-learning (DL) based. Deep learning models consist of context encoder and tag decoder. The context encoder applies a DL model to produce a sequence of contextualized token representation used as input for tag decoder which assign entity class for each token in sequence. For a comprehensive survey see (Li et al., 2020). In most entity extraction systems the EE task is explicitly (or for some DL models implicitly) separated into NER an NEN tasks.

Feature-based approaches solve the NER task as a sequence markup problem by applying such feature-based models as Hidden Markov Models (Okanohara et al., 2006) and Conditional Random Fields (Lu et al., 2015). The downside of such models is the requirement of extensive feature engineering. Another method for NER is to use DL models (Ma & Hovy, 2016) (Lample et al., 2016). This models not only select text spans containing named entities but also extract quality entity representations which can be used as input for NEN. For example in (Ma & Hovy, 2016) authors combine DL bidirectional long short-term memory network and conditional random fields.

Main approaches for NEN task are: rule-based (D'Souza & Ng, 2015) (Kang et al., 2013), feature-based (Xu et al., 2017a) (Leaman et al., 2013) and DL methods (Li et al., 2017a) (Luo et al., 2018b) and their different combinations (Luo et al., 2018a). Among DL approaches a popular way is to use distance metrics between entity representations (Ghiasvand & Kate, 2014) or ranking metrics (Xu et al., 2017a) (Leaman et al., 2013). In addition to ranking tasks DL models are used to create contextualized and more representative term embeddings. This is done with a wide range of models: Word2Vec (Mikolov et al., 2013), ELMo (Peters et al., 2018), GPT (Radford et al., 2018), BERT (Devlin et al., 2018). The majority of approaches combine several DL models to extract context-aware representations which are used for ranking or classification using a dictionary of reference entity representations (Ji et al., 2020).

The majority of modern medical EE systems sequentially apply NER and NEN. Considering that NER and NEN models themselves are often multistage the full EE systems are often complex combinations of multiple ML and DL models. Such models are hard to train end-to-end and if the NER task fails the whole system fails. This can be partially mitigated by simultaneous training of NER and NEN components. In (Durrett & Klein, 2014) a CRF model is used to train NER and NEN simultaneously. In Le et al. (2015) proposed a model that merged NER and NEN at prediction

time, but not during training. In Leaman & Lu (2016) proposed semi-Markov Models architecture that merged NER and NEN both at training and inference time. Even with such merging of NER and NEN both tasks were present in the pipeline which proves problematic in difficult cases with multi-word entities or single entities with non-relevant text insertions.

A number of deep-learning EE models (Strubell et al., 2017), (Li et al., 2017b), (Xu et al., 2017b), (Devlin et al., 2018), (Cui & Zhang, 2019) do not split the EE task into NER and NEN implicitly and use a single linear classification layer over token representations as the tag decoder. Our model is mostly identical to the model described in (Devlin et al., 2018) with the difference that instead of using a contexualized representation of each token to classify it as an entity we use the representation of the whole text to extract all entities present in the text at once.

Supervised training of EE systems requires large amount of annotated data, this is especially challenging for domain-specific EE where domain-expert annotations is costly and/or slow to obtain. To avoid the need of hand-annotated data various weakly-supervised methods were developed. A particular instance of weak annotation is distant annotation which relies on external knowledge base to automatically label texts with entities from KB (Mintz et al., 2009), (Ritter et al., 2013), (Shang et al., 2018). Distant supervision can been applied to automatically label training data, and has gained successes in various natural language processing tasks, including entity recognition (Ren et al., 2015), (Fries et al., 2017), (He, 2017). We use distant annotation in this paper to label our train and test datasets.

## 3 DATA

### 3.1 ELECTRONIC HEALTH RECORDS DATASETS

In this work we used two proprietary Russian language EHR datasets, containing anonymized information. First one contains information about 2,248,359 visits of 429,478 patients to two networks of private clinics from 2005 to 2019. This dataset does not contain hospital records and was used for training the model. The second dataset was used for testing purposes and comes from a regional network of public clinics and hospitals. Testing dataset contains 1,728,259 visits from 2014 to 2019 of 694,063 patients.

### 3.2 MEDICAL KNOWLEDGE BASE

We used UMLS as our medical KB and a source of medical entity dictionary for this paper. A subset of UMLS, Medical Dictionary for Regulatory Activities (MedDRA) was used to obtain translation of terms to Russian language. After merging the synonymous terms we selected 10000 medical entities which appeared most frequently in our training dataset. To find the terms we used a distantly supervised labelling procedure as described in next section. To increase the stability of presented results we decided to keep only terms that appear at least 10 times in the test dataset reducing the total number of entities to 4434. Medical terms were grouped according to UMLS taxonomy, statistics for group distribution are shown in Table 1.

### 3.3 DISTANT SUPERVISION LABELING

Combining an EHR dataset and a list of terms from medical KB we used a simple rule-based model for train and test datasets labeling. The exact procedure for each record was as follows:

- Input text was transformed to lower case, all known abbreviations where expanded, and all words were lemmatized using pymorphy2 (Korobov, 2015)

- We selected all possible candidates using sliding window with lengths from 1 to 7 words

- All possible candidates where compared to all possible synonims of medical entities

- Exact matches between candidate and medical terms from KB where considered to be positive cases.

Table 1: Medical entity group statistics

| Entity group | Total terms | Instances in train | Instances in test |
|---|---|---|---|
| Diagnostic Procedure | 157 | 654.222 | 301.279 |
| Disease or Syndrome | 1307 | 2.204.636 | 2.318.028 |
| Finding | 475 | 2.137.647 | 1.287.896 |
| Injury or Poisoning | 168 | 230.543 | 159.913 |
| Laboratory Procedure | 141 | 891.110 | 380.129 |
| Neoplastic Process | 212 | 132.732 | 117.311 |
| Pathologic Function | 231 | 600.567 | 288.433 |
| Pharmacologic Substance | 324 | 474.033 | 263.762 |
| Sign or Symptom | 368 | 3.912.937 | 2.279.892 |
| Therapeutic or Preventive Procedure | 352 | 287.533 | 218.826 |
| All other groups | 699 | 3.527.664 | 1.821.642 |

## 4 MODEL

In this paper we used a RuBERT model pretrained on general russian texts (Kuratov & Arkhipov, 2019) and further pretrained on electronic health records. A linear classification layer with 10000 outputs was added as the last model layer (Fig 1.). This layer was initialized with weights from normal distribution with mean=-0,1 and std=0,11 to have at the start of training a low prediction probability for all classes.

We trained our model with binary crossentropy loss and Adam optimizer (Kingma & Ba, 2014) with learning rate 0.00001 making one pass over training dataset with training batches of size 20. To speed up training we used dynamic class weightings, classes not present in the current batch were given less weight compared to classes present. Model architecture is shown on Figure 1.

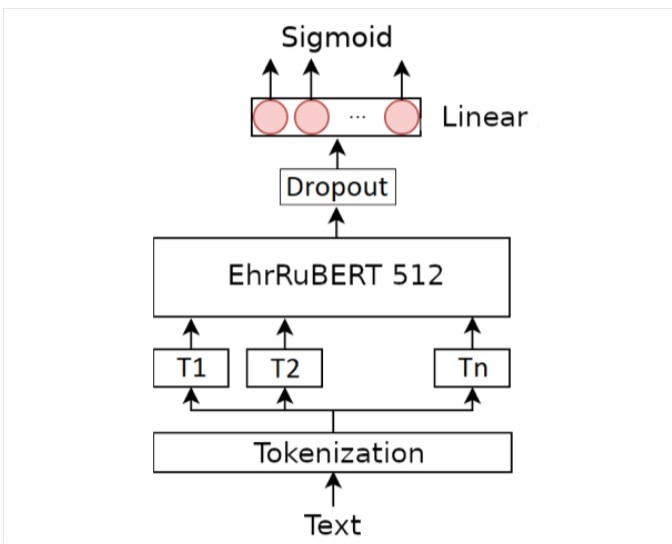

Figure 1: Model architecture

## 5 RESULTS

### 5.1 DISTANT LABELS

Using distantly-generated labels we calculated the recall of our model on the test dataset. Our expectations were that with enough training examples the model should achieve recall close to 1.

We found that for some categories for like 'Pharmacologic Substance' the model did not learn to correctly find entities even with a large enough number of training instances. The relation between number of training examples an recall on the test for entity classes: Sign or Symptom, Finding and Pharmacological Substance, and for all entities is shown on Fig 2.

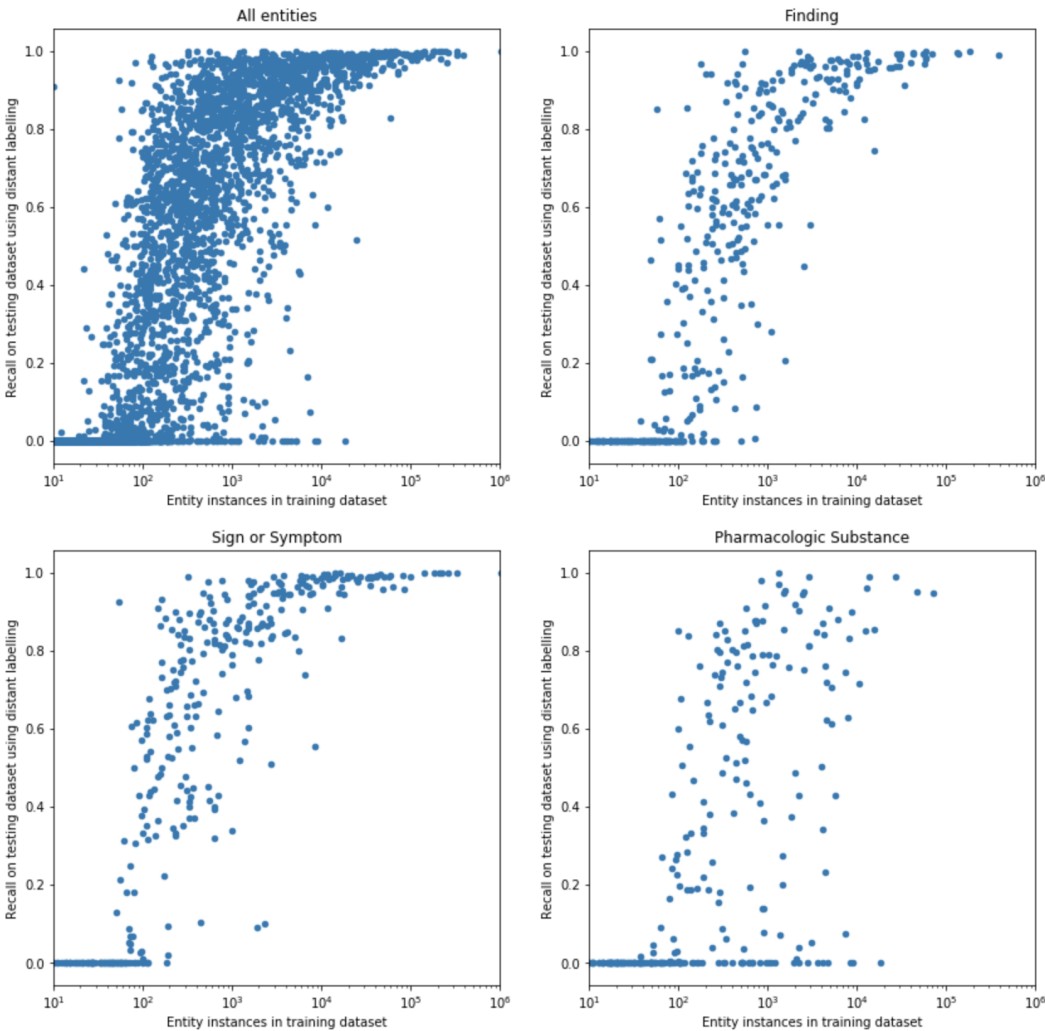

Figure 2: Relation between number of training examples and recall on test data

As can be seen in Table 2 the number of training examples needed to achieve given recall differs between classes with some classes needing noticeably more examples. There could be numerous sources of such difference: tokenization, number of synonims, difficult context (substances are often as encountered lists mixed with dosages) and others. Even for the harder classes fifty thousand examples are enough to find nearly all distant labels

## 5.2 HUMAN LABELING

A major disadvantage of our labelling procedure is its incompletness. Any slight change of known term, a typo or a rare abbreviation will lead to missed entities. This makes estimating the precision of the model impossible with such labels. To compare our model with a human benchmark we randomly selected 1500 records from the testing dataset for hand labelling by a medical practitioner with 7 years of experience. These records where labeled for 15 most common entities in train dataset. After labeling we further analysed the cases where the model disagreed with human annotator by splitting all instances into following cases:

Table 2: Recall on test dataset

| | Examples in train dataset | | | | | | | |
| | >0 | | >500 | | >2.500 | | >50.000 | |
| **Entity group** | **#** | **recall** | **#** | **recall** | **#** | **recall** | **#** | **recall** |
|---|---|---|---|---|---|---|---|---|
| Diagnostic Procedure | 157 | 0.39 | 63 | 0.76 | 29 | 0.86 | 6 | 1.0 |
| Disease or Syndrome | 1307 | 0.33 | 338 | 0.83 | 138 | 0.91 | 10 | 0.99 |
| Finding | 475 | 0.42 | 167 | 0.81 | 80 | 0.93 | 10 | 0.99 |
| Injury or Poisoning | 168 | 0.33 | 26 | 0.88 | 8 | 0.96 | 1 | 0.98 |
| Laboratory Procedure | 141 | 0.44 | 75 | 0.71 | 35 | 0.86 | 2 | 0.99 |
| Neoplastic Process | 212 | 0.18 | 28 | 0.78 | 12 | 0.90 | 0 | - |
| Pathologic Function | 231 | 0.32 | 61 | 0.81 | 25 | 0.92 | 4 | 0.98 |
| Pharmacologic Substance | 324 | 0.26 | 104 | 0.53 | 40 | 0.60 | 1 | 0.95 |
| Sign or Symptom | 368 | 0.55 | 159 | 0.87 | 87 | 0.95 | 14 | 0.99 |
| Therapeutic Procedure | 352 | 0.30 | 74 | 0.82 | 24 | 0.86 | 0 | - |
| All entities | 4434 | 0.36 | 1344 | 0.79 | 608 | 0.89 | 62 | 0.98 |

Table 3: Discrepancies between human and model labeling

| | | | | correct | | | |
| | | both | | human | | model | |
| **Entity** | **Examples** | **tp** | **tn** | **tp** | **tn** | **tp** | **tn** |
|---|---|---|---|---|---|---|---|
| Pain (Sign or Symptom) | 1.011.037 | 400 | 1079 | 5 | 5 | 0 | 11 |
| Frailty (Sign or Symptom) | 390.422 | 41 | 1448 | 3 | 2 | 1 | 5 |
| Coughing (Sign or Symptom) | 328.236 | 118 | 1358 | 17 | 1 | 0 | 6 |
| Complete Blood Count (Laboratory Procedure) | 326.281 | 22 | 1468 | 2 | 3 | 0 | 5 |
| Rhinorrhea (Sign or Symptom) | 261.519 | 62 | 1427 | 5 | 1 | 0 | 5 |
| Evaluation procedure (Health Care Activity) | 253.766 | 39 | 1455 | 0 | 1 | 0 | 5 |
| Illness (Disease or Syndrome) | 245.042 | 90 | 1301 | 4 | 6 | 0 | 99 |
| Headache (Sign or Symptom) | 222.656 | 83 | 1408 | 7 | 0 | 0 | 2 |
| Ultrasonography (Diagnostic Procedure) | 218.481 | 40 | 1448 | 1 | 0 | 0 | 11 |
| Discomfort (Sign or Symptom) | 211.075 | 13 | 1477 | 0 | 4 | 0 | 6 |
| Discharge, body substance (Sign or Symptom) | 184.181 | 4 | 1486 | 1 | 0 | 0 | 9 |
| Nasal congestion (Sign or Symptom) | 183.886 | 20 | 1475 | 2 | 0 | 0 | 3 |
| Abdomen (Body Location) | 176.965 | 27 | 1465 | 0 | 0 | 0 | 8 |
| Urinalysis (Laboratory Procedure) | 171.541 | 14 | 1485 | 0 | 0 | 0 | 1 |
| Infection (Pathologic Function) | 154.753 | 13 | 1464 | 2 | 0 | 13 | 8 |

- both correct - model and annotator found the term (true positive)

- both correct - model and annotator did not find the term (true negative)

- model correct - found term missed by annotator (model true positive)

- model correct - did not find erroneously annotated term (model true negative)

- model wrong - non-existing term found (human true negative)

- model wrong - existing term missed (human true positive)

From the results presented in Table 3 we can conclude that our model in general extracts most frequent entities with human-level quality. Large number of annotator errors for entities 'Illness' and 'Infection' stem from their occurrence in multiple acronyms and so are easily missed. Large number of model errors in case of 'Coughing' are due to a single term synonym that was almost completely absent from train dataset and present in test dataset.

Table 4: Examples of generalization by the entity extraction model

| Original text | Extracted entity | Comments |
|---|---|---|
| leakage of urine into the diaper | Urinary incontinence | A correct entity is extracted even though the form used is not in the list of synonims from the knowledge base. |
| prickling pains with feeling of pressure in the heart | Pain in the heart region | Correct entity extraction in with extra words inside the entity span. |
| complaints of pain pain in the lumbar joint | Pain in lumbar spine | Using the word joint as an anchor the model correcctly selected the term 'Pain in lumbar spine' instead of closely related terms 'Low back pain' or 'Lumbar pain'. |
| complaints of pain in the abdomen, right hypochondrium | Right upper quadrant pain ... | The entity is extracted correctly even with body location 'Abdomen' in the middle of the phrase. |
| complaints of trembling fingers when excited | Shaking of hands | Correct extraction of unusual entity form. |
| blood pressure occasionally rises | Increase in blood pressure; Blood pressure fluctuation | Using the word 'occasionaly' the model in addition to general entity 'Increase in blood pressure' successfully extracts a correct more specific entity 'Blood pressure fluctuation'. |
| a child on disability since 2008 after a cytomegalovirus infection with damage to the heart, hearing organs, vision, central nervous system | Central nervous system lesion ... | Model correctly connects the word damage with term central nervous system even though they are separated by several words and punctuation marks and extracts the corresponding entity. |
| intercost neurlgia | Intercostal neuralgia | Typos ignored when extracting the correct entity |

## 5.3 EXAMPLES

In this section we selected several examples of model generalising in difficult cases. In Table 4 we provide the original text translated into English and the extracted entity also in English form with our comments.

## 6 CONCLUSION

In this paper we show that a single transformer model can be used for one-step medical entity extraction from electronic health records. This model shows excellent classification quality for most frequent entities and can be further improved by better language pretraining on general or in-domain texts, hyperparameter tuning or applying various ensembling methods. Not all entity classes are easily detected by model. Some classes like 'Farmacologial Substances' are noticeably harder to classify correctly. This can be due to number factors including differences in context, number of synonims and the difference between train and test dataset.

We have shown that 50.000 training examples are enough for achieving near perfect-recall on automatic labels even for hard classes. Most frequent entities, with more that 150.000 training examples

are classified with human-level quality. We did not explicitly search for lower limit of training examples needed to achieve such quality so it can be substantially lower. Also we showed that such quality is achieved even when using testing dataset which greatly differs in entity distribution, geographic location and institution types (clinic vs hospital) from training dataset. This implies the model ability to generalize to new, unseen and difficult cases after training on a limited variety of reference strings for each entity.

The number of errors made by human annotator highlights the hardships that medical entity annotation poses to humans, including the need to find inner entities in complex and abbreviated terms. The markup of the complete medical entities vocabulary is also problematic due to both a large number of entities possibly present in each record and to the fact that some entities are exceedingly rare. Less than half of training entities appearing at least 10 times in the testing dataset. A complete markup of such infrequent entities is not really feasible as it would involve looking through an extremely large number of records to get a reasonable number of entity occurrences.

The proposed distantly-supervised method can be used to extract with human-level accuracy a limited number of most frequent entities. This number can be increased by both improving the quality of the model and by adding new unlabeled examples. Distantly supervised entity extraction systems made in line with our model can be used for fast end resource-light extraction of medical entities for any language. While currently limited to a small vocabulary of common terms such systems show big promise in light of increasingly available amounts of medical data.

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
