# OpenReview forum: "Distantly supervised end-to-end medical entity extraction from electronic health records with human-level quality"
_ICLR.cc/2021/Conference — Reject_

### Official Review · AnonReviewer2 · 2020-10-27
**NER and entity linking with distant supervision for Russian text using BERT as a classifier**

**Rating:** 5
**Confidence:** 5

**Review:**

This paper introduces an end-to-end task that identifies medical
entities and links them to UMLS concepts for Russian biomedical
text. The paper uses distant supervision to identify the entities with
their corresponding concepts and uses a Russian pre-trained BERT model
to perform the task. The most frequent 10K medical concepts are
selected and the task is treated as a classification task.

The strength of	this paper is that it is probably (one of) the first
to address NER and entity linking for biomed concepts in Russian with
a straightforward BERT model. The main weakness of the paper stems
from the way the training/validating datasets are created.

The dataset is extracted from EHR notes	and the	labeling is done by a
simple rule-based model that involves exact matching. I'm not
convinced that this is going to have general coverage, and, hence, the
model and the validation are biased to the entities for which exact
matching appear more often. The validation includes also a manually
annotated portion (1.5K records) labelled only for the top 15 most
common entities. I dind't follow the numbers in Table 3 too well (not
clear what column # represents). The recall once a large amount of
text is used for training seems quite high (high 90s). However, I
haven't seen a discussion on the precision, which, from my experience
with English text, seems to be more problematic.

I didn't fully understand how the entity spans are detected. I think 1
could include more details with a concrete example. If you think about
it, that figure is so general that it could be part of any paper
(minus the specialized version of BERT).

I think	the paper is for the most part well written. The
validation/evaluation is rather weak,	with some stats	missing.

My take	is that	this paper does	not have a high	degree of novelty and
it is also rather niche topic, as such, I think it's better suited for
a more specialized venue such as the BioNLP workshops collocated with
ACL conferences.

---

### Official Review · AnonReviewer1 · 2020-10-28
**Task and model not clearly defined; lack of comparisons with baselines**

**Rating:** 4
**Confidence:** 4

**Review:**

This paper presents a distantly supervised approach to extracting medical entities from EHR. The authors use (a translated subset of) UMLS to match the unannotated corpus to generate distant training data, and then feed the training data into a BERT-based model to jointly perform named entity recognition (NER) and named entity normalization (NEN).

I have the following major concerns about this submission:

1. Based on my understanding (according to Figure 1 and Table 4), the proposed approach is performing multi-label text classification instead of entity extraction. The NER task aims to detect token spans in the text that represent an entity. It is an interesting proposal to simultaneously do NER and NEN, but you still need to output the location/boundary of the extracted entity in the text. Without this information, it would be difficult to perform downstream tasks like relation extraction.

2. Several details of generating distant supervision are missing.
- How do you expand "all known abbreviations"? Acronyms are a large source of noise when generating distant supervision. If you blindly match all abbreviations, there will be many false-positive training samples.
- How do you deal with the conflict during matching? For example, given a word sequence w1w2w3, one entity in the dictionary can match w1w2, while another can match w2w3. What will be your matching result?
- Shang et al. [1] point out that matching all possible synonyms of a word will also generate noise. They suggest that only when the canonical name of a word appears in the text, its synonym will be considered. Please consider comparing their strategy with yours.

3. The paper only shows the results of its own approach. Several benchmark approaches need to be compared, such as Swellshark [2] and AutoNER [1].

[1] Shang et al. Learning named entity tagger using domain-specific dictionary. EMNLP'18.

[2] Fries et al. SwellShark: A Generative Model for Biomedical Named Entity Recognition without Labeled Data. arXiv'17.

Typos: "where" -> "were"; "synonim" -> "synonym"

---

### Official Review · AnonReviewer3 · 2020-10-29
**Distantly supervised end-to-end medical entity extraction review**

**Rating:** 4
**Confidence:** 4

**Review:**


This paper presents a model that performs Medical Entity Extraction in a way that 1) allows to generalise better certain types of entities, resulting in a better performance and 2) can achieve human-like quality despite relying on a distantly supervised training.

PRO's:
- State of the art is compelling
- The authors are clear in stating the contributions of their work, and how the model has been implemented.
- The topic of labelling medical documents is highly relevant in these covid times: being able to automatically process medical documents is a challenge that could help doctors and health experts to deal with the huge amount of information that is being publish daily.
- Include experiments on Russian, good for a domain where most of the research is focusing on English.

However, imho the contributions stated are not significant enough for the paper to be accepted for publication. In particular:
1) The model architecture does not bring any novelty. (It is basically BERT+Linear layer). The pretrained model helps improving the labelling task, but this has been already observed in many other NLP domains and papers.
2) Distant supervision is attractive, as we don't rely on expensive labelled data, but the performance analysis is limited to a few (most-frequent) classes. In a domain like health, the long tail of entities can be crucial so this techniques might not be suitable for some use cases.
3) It is too risky to claim human-like performance, when only one expert annotator has been involved.

In addition to this, it would be nice if the authors can address the following comments:
- In related work you claim that "Instead of using a contexualized representation of each token to classify it as an entity we use the representation of the whole text to extract all entities present in the text at once". Can you clarify how this is achieved?
- In addition to the number of classes in train versus test datasets, could you elaborate a bit more on other differences, e.g. average length of the documents, presence of some structured data within the documents (e.g. template)...
- Is the fact that you used Russian datasets making your challenge harder? E.g would you be able to better leverage UMLS if you run similar experiments in medical records in English?
- "we decided to keep only terms that appear at least 10 times in the test dataset reducing the total number of entities to 4434" -> Ideally, you should not rely on any clues or signals from the test dataset, so you don't bias your model toward specific categories present in the test data. I would suggest to repeat experiment without this step and report what happens with this less frequent classes too.
- Section 4 is to scarce, more details about the model and the training should be included, for example: how did you pretrained RuBERT on electronic health records? E.g. how much data was available for this domain, for how many epochs you trained... Also, How do you think this is affecting the end2end performance of your model?
- As stated earlier, one annotator is not enough for a proper human evauation. I suggest to be more exhaustive in this study and bring in some more experts to annotate the text, so we can calculate agreement score.
- Please include more details when you refer to model's mistakes. e.g. "Large number of model errors in case of ’Coughing’ are due to a single term synonym that was almost completely absent from train dataset and present in test dataset." -> Indicate the term
- Table 4: How can you quantify the improvement achieved by these entity extraction generalizations? The distant supervision is not able to capture those advanced use cases when labelling entites so, do you have ideas of how many of those you might be ignoring during your evaluation?
- 150.000 training examples to achieve human-quality: Are you sure this is good enough? What happens when the number of available classes increase?

---

### Official Review · AnonReviewer4 · 2020-10-29
**Distantly supervised end-to-end medical entity extraction from electronic health records with human-level quality**

**Rating:** 3
**Confidence:** 4

**Review:**

Summary:
This paper proposes a method to do medical entity extraction from HER data by fine-tuning a transformer model pretrained on a large EHR dataset.  The model combines a two-step process of NER and NEN into a single step on a multi-label classification task by distantly supervised training. The main contribution of this paper is to exploit a single transformer model to perform NER and NEN for HER data simultaneously by using the representation of EHR for a single multi-label classification task.   Empirical studies are performed to show the expected recall.

Pros:
1.	The paper introduces distant annotation label data to avoid domain-expert costly annotations.
2.	Apply pre-trained transformer-based model to finetune the proposed model in EHR data to do medical EE.
3.	Large EHR data pre-processing

Cons:
1.	The paper does not give the performance comparison with the state-of-the-art EE model.
2.	The originality and significance of this paper is not enough, as it applies RuBERT on EHR data to do medical EE.
3.	It is better to give additional evaluation metrics, such as precision and F score.

Minor comments:
1.	In the last sentence of the second paragraph in Related work, it should be " into NER and NEN tasks. " instead of " into NER an NEN tasks. ".
2.	Brief introduction about RuBERT in model section.
3.	There are some typos and grammatic error.

---

### Decision · Program_Chairs · 2021-01-07
**Final Decision**

**Decision:**

Reject

**Comment:**

This paper tackles an important problem and includes experiments on a new domain (Russian documents vs English documents). Unfortunately, all reviewers agree that this paper lacks novelty for publication in its current state. Additional details and clarifications to the proposed approach, notably through a more thorough performance analysis, would improve the significance of the paper.